# Genetic Strategies for Enhancing Rooster Fertility in Tropical and Humid Climates: Challenges and Opportunities

**DOI:** 10.3390/ani15081096

**Published:** 2025-04-10

**Authors:** Jiraporn Juiputta, Vibuntita Chankitisakul, Wuttigrai Boonkum

**Affiliations:** 1Department of Animal Science, Faculty of Agriculture, Khon Kaen University, Khon Kaen 40002, Thailand; jiraporn.ju@kkumail.com (J.J.); vibuch@kku.ac.th (V.C.); 2Network Center for Animal Breeding and Omics Research, Khon Kaen University, Khon Kaen 40002, Thailand

**Keywords:** genome selection, semen quality, native chickens, heat stress, GWAS

## Abstract

Native chickens are important for both economic reasons and for preserving genetic diversity. Hot and humid weather affects the quality of semen and the breeding ability of male chickens. Genetic selection using genomic technology can help solve this problem by improving the accuracy and shortening the selection time. By comparing various genomic selection models and genome-wide association studies (GWASs), researchers can identify genes related to semen quality and heat tolerance. This helps breeders and researchers to develop effective genetic selection plans for future generations of chickens.

## 1. Introduction

The poultry industry faces many challenges, including disease control, product quality maintenance, and ensuring appropriate production costs. This is particularly important considering the evolving needs of consumers [1]. Consumers are currently looking for high-quality and nutritious products as people become more health conscious. Certain foods reduce the risk of diseases such as heart disease, diabetes, and obesity [2]. Therefore, chicken, which is white meat, is becoming more popular, because it has less fat, lower cholesterol, and a higher protein content than red meat. Furthermore, it is an affordable animal protein source [3]. Potue et al. [4] pointed out that some Thai native chickens are hybrids, resulting from crossbreeding between commercial broiler chickens and pure Thai native chickens, which have the potential to be a functional food because they have a positive effect on burning fat. Furthermore, the high glutamic acid content of native chickens contributes to their delicious mellow taste [5,6]. Native Korean chickens have higher levels of bioactive compounds than commercial broilers [7]. Consequently, native chicken has become increasingly popular. In 2024, it was found that the consumption of indigenous chicken meat has increased, increasing by about 89% in 2018 compared to that in 2005 in Korea, and many other countries, including Thailand, Indonesia, and Italy, have reported an increase in consumption [3,8,9]. Moreover, native chickens are resistant to various climates and adapt well to their environments [10], making them ideal for sustainable agriculture. Native chickens also have unique genetic traits and are important biological resources for protecting their genetic diversity [11]. This will help to create future breeds that are more resistant to environmental damage and can live indefinitely. However, of the 1499 breeds surveyed, 28% had a conservation status ranging from endangered to critically endangered, and 3.4% were extinct. Commercial breeding focuses primarily on hybrids [12]. This poses the risk of losing the genetic diversity of valuable native chickens [13].

Developing strategies to increase productivity and reduce costs is an important goal for livestock farmers. In the past, genetic modification was high to increase the growth rate of chickens to produce more chicken meat quickly, resulting in lower production costs and increased profits for farmers. However, focusing on a single trait may negatively affect the other important traits. Fertility traits significantly influence chick production [14]. This is especially true in roosters, where a single rooster can mate with multiple hens, thereby directly influencing the fertilization rate and number of chicks born. In addition, roosters play a role in transmitting important genetic traits such as strength, good growth, and disease resistance, which help to strengthen the breed in the long term [15]. Semen traits are among the most important economic traits in the chicken breeding industry. Semen production reflects the reproductive efficiency of chickens, which directly affects their production and reproductive abilities [16]. In addition, roosters can produce sperm continuously and for a longer period than the egg production lifespan of hens, making them economically valuable. Maintaining and enhancing the reproductive health of male chickens plays a crucial role in lowering costs and boosting production efficiency, ultimately contributing to the overall success of poultry production.

However, hot and humid weather conditions are important factors affecting poultry production, particularly heat stress (HS), which is likely to be exacerbated by the current increase in temperature [17]. These conditions expose chickens to temperatures that surpass their tolerance. Although native chickens have excellent environmental adaptation, native chickens can best control their temperature in the range of 21–26 °C, but when the temperature reaches 32 °C, it will cause HS [18,19], which results in higher mortality rates, lower production, and significantly reduced semen quality. Although this problem can be solved by supplementing or managing food [20,21,22], these approaches are only temporary and increase production costs. Therefore, genetic improvements are needed to improve traits, such as high-quality semen and heat tolerance [23]. However, fertility and heat tolerance in chickens have low heritability because environmental factors have a greater effect than genetics [24,25]. Currently, traditional selection is declining in popularity, owing to its lengthy process and low accuracy, as it relies solely on phenotypes and pedigree for prediction, whereas genomic selection (GS) is continuously gaining popularity. Several studies have demonstrated that the integration of GS enhances the accuracy of selection, often exceeding 0.5 for fertility-related traits, and reduces the generation interval by 30–50%. This transition allows for the quicker identification of superior sires based on genomic estimated breeding values (GEBVs), shortening traditional selection cycles from 1.5 to 2.0 years to under a year [26,27,28]. Therefore, genetic selection methods utilizing genetic markers have been developed to address this issue. These methods enhance the speed and accuracy of selecting animals with desired genetics.

Therefore, the goal of the present study is to analyze previous genetic studies that investigated genetic improvements in fertility traits and heat tolerance in male chickens. This will also reveal different genetic selection models that will enable faster and more accurate chicken breed selection for future production and breed development.

## 2. Effects of HS on Fertility Traits of Roosters

HS is a major problem in tropical animal husbandry, occurring when chickens cannot maintain body heat balance. The severity of the disease is determined by factors including environmental temperature and humidity, as well as breed and metabolic rate in the chickens themselves [29]. The Temperature and Humidity Index (THI) is often used as an indicator to assess the level of HS in livestock. Juiputta et al. [23] reported that a THI of 78 reduced semen quality, especially sperm concentration, in Thai native chickens (Figure 1). However, the thermoneutral zone of native chickens varies significantly among breeds and has not yet been clearly established by breed and environment. HS is divided into two main types that affect the reproductive system: (1) Acute HS is caused by a rapid increase in temperature in a short period; for example, experiments that exposed breeds to 38 °C for 4 h found that sperm stem cells deteriorate and cell death increases. As a result, spermatogenesis was impaired, and semen quality decreased [30]. (2) Chronic HS, for example, when the roosters were at 38 ± 1 °C and the humidity was 55–65% RH for 3 consecutive days, this led to lower sperm concentration and viability and higher malondialdehyde (MDA) levels [31].

In chickens, the testicles and epididymis are located inside the abdomen, where they remain fixed in position. Therefore, testicular temperature regulation relies on a cooling mechanism different from that of mammals. The pampiniform plexus is responsible for heat exchange between the arteries and veins, which lowers the temperature of the blood before it enters the testicles. This mechanism helps to maintain the optimum temperature for sperm formation and prevents thermal damage.

Under normal conditions, the Gonadotropin-releasing hormone (GnRH) stimulates the posterior pituitary gland to release follicle-stimulating hormone (FSH) and luteinizing hormone (LH) (Figure 2). These hormones cause testosterone production in the testicles, which in turn normalizes spermatogenesis and sperm production [32]. In addition, the balance between reactive oxygen species (ROS) and antioxidants under normal conditions helps proteins in seminal plasma function, resulting in good sperm motility, viability, and semen quality. In contrast, HS affects the hypothalamus, releasing Gonadotropin-inhibitory hormone (GnIH), which inhibits the release of GnRH, FSH and LH, resulting in reduced testosterone production, spermatogenesis, and a lower sperm concentration [33]. Moreover, HS causes an imbalance in which ROS levels are higher than that of antioxidants, resulting in oxidative damage to cells, including DNA damage, lipid degradation, and acrosome damage. Destruction of sperm mitochondria results in cell death (apoptosis), reduced sperm motility and viability, and low semen quality [34].

## 3. Genetic Improvement to Improve Production Efficiency in HS

In animal breeding for HS resistance, the interaction between genotype and environment (G × E interaction) should be considered. The G × E principle states that different genotypes perform differently under different environments. In quantitative genetics, if the genetic correlation between the same trait in two environments is less than 0.80, it is considered to display G × E interaction [34]. Thus, changes in the environment, such as higher temperatures or humidity, can change the fitness of animals in different areas [35]. The THI has been used to create a genetic model showing how temperature and humidity affect genetic traits and animal performance. This can help to predict traits influenced by both genetic and environmental factors. Bohlouli et al. [36] stated that a model that includes the THI is better at predicting traits that change because of environmental stress, including monthly egg production [37], growth [38], and semen quality [23], and can more accurately show how genotype and the environment work together. Figure 3 shows that higher THI values reduce genetic and environmental variability and heritability. This indicates that HS negatively affects both genetic adaptation and selection. However, heat tolerance and genomic estimated breeding value (GEBV) in the fertility traits of roosters have not been considered together. This aspect should be considered in future studies. This model can help to select animals that can maintain high production, even in challenging environments.

Future research integrating heat tolerance traits with genomic selection is crucial for fully addressing genotype-by-environment (G × E) interactions in practical breeding scenarios. To improve this integration, future studies should focus on the following: (1) How advancing genomic selection requires the identification of reliable genetic markers associated with heat tolerance. Genome-wide association studies (GWASs) and whole-genome sequencing can help to pinpoint single nucleotide polymorphisms (SNPs) linked to heat resilience traits, such as body temperature regulation, feed intake stability, and oxidative stress response. (2) Advanced statistical models, such as reaction norm and random regression models, can capture G × E interactions more accurately. These models allow for genetic evaluations under different thermal environments, identifying sires that perform consistently across varying Temperature–Humidity Index (THI) levels. (3) Future studies could integrate high-density genomic data with long-term environmental records to better characterize these interactions in semen traits. Additionally, incorporating whole-genome sequencing and gene–environment association analyses may provide deeper insights into the genetic mechanisms underlying heat stress resilience in native chickens. Such advancements could contribute to more precise genetic selection strategies, ultimately improving productive performance and adaptability in poultry breeding programs [35,36].

Heat stress poses a significant challenge to poultry production, particularly in tropical and subtropical regions. In native roosters, genetic evaluation models traditionally focus on production traits such as growth rate, body weight, and feed efficiency. However, integrating heat tolerance traits into these models is crucial for improving resilience and overall productivity under high-temperature conditions. One approach to incorporating heat tolerance is through the inclusion of physiological and behavioral indicators, such as body temperature regulation, respiratory rate, and heat-shock protein expression, as additional traits in multi-trait genetic models. These traits can be genetically correlated with production traits, allowing for simultaneous selection. Another strategy involves using the Temperature–Humidity Index (THI) as an environmental covariate in reaction norm models. This method helps to estimate genotype-by-environment interactions, allowing for the identification of roosters that maintain stable production performance under varying heat stress conditions. By modeling genetic parameters across different THI levels, breeders can select individuals with higher resilience while maintaining desirable production traits. The Temperature–Humidity Index (THI) is an effective measure for assessing the impact of heat stress, as it comprehensively assesses the impact of temperature and humidity and is easy to collect data on compared to other indicators such as body temperature, respiratory rate, heat loss, and stress-related hormone levels. Therefore, using THI in genetic models and specifying it as a covariate or random effect can better capture the interaction between genotypes and the environment. Previously, it was found that when THI reached 76, animals lost growth traits, and THI at 74 reduced egg production, indicating the impact of heat stress on animal performance [37,38]. Integrating THI in genetic analysis can increase the accuracy of genetic value assessment and help to create selection strategies that respond to heat stress effectively.

Genomic selection also plays a vital role in integrating heat tolerance. By identifying the genetic markers associated with thermoregulation and incorporating them into selection indices, breeders can improve the genetic potential for heat resilience without compromising growth and reproductive efficiency [39]. Additionally, the use of non-linear models, such as Gompertz and logistic functions, can help to capture the dynamic effects of heat stress on growth patterns and optimize breeding strategies accordingly. In summary, incorporating heat tolerance traits into genetic models for production traits in native roosters requires a multi-faceted approach, including multi-trait analysis, reaction norm models, genomic selection, and non-linear modeling. This integration enhances the ability to breed roosters that are both productive and resilient to heat stress, ensuring sustainability in tropical poultry production. 

## 4. Genomic Selection and Model Prediction

Genomic selection (GS), using data from genetic markers that span the entire genome to improve the accuracy of GEBVs and accelerate the selection process, has become an important technique in animal selection. This technology reduces the limitations of traditional breed selection, which relies solely on pedigree and phenotypic data, as GS can select high-potential animals at an early age before displaying phenotypic traits. This reduces the generation interval and significantly reduces the cost of phenotypic testing [40]. Advances in single-nucleotide polymorphism (SNP) genotyping technology have made it possible to analyze chicken genomic data more accurately and in detail. As a result, GS has been widely applied in both broiler and laying hens to improve key economic characteristics such as growth rate, feed efficiency, and egg production [26,41,42]. Although GS is highly effective, the accuracy of GEBV prediction depends on many factors, such as the density of SNP markers, the traits studied, heritability, and the statistical model used in the analysis. Recently, models of genomic GS have been developed to increase the accuracy of chicken breeding selection. These models have been improved from the basic GBLUP to ssGBLUP, which combines pedigree data, the Bayesian model that gives different SNP weights, the MTGBLUP for multiple traits, the RR-GBLUP that fits the traits that are controlled by multiple genes, and the WGBLUP, which increases accuracy by weighting SNPs according to their influence. These developments allow for a more efficient and accurate selection of chicken breeds, which will be discussed in more detail later.

### 4.1. Genomic Best Linear Unbiased Prediction (GBLUP)

The GBLUP method uses genomic data to evaluate an individual’s genetic value. This method enables a higher accuracy than traditional methods that use only pedigree data because it reflects the degree of genomic similarity more directly [43]. Therefore, this method is highly effective in predicting the genetic value of traits regulated by many genes and in large populations [44]. However, GBLUP has a major limitation in assuming that all SNPs have equal impact, which prevents the proper identification of major effect SNPs, which affects the prediction accuracy of traits controlled by a limited number of QTLs. In addition, GBLUP is not suitable for analyzing complex traits because it cannot handle phenotypic data with non-linear relationships. To solve these problems, researchers have developed advanced models, such as Bayesian models that can weight SNPs differently according to their impact [45,46]; WGBLUP, which uses SNP weighting to improve accuracy, especially when high impact SNPs are known [47]; and deepGBLUP, which integrates deep learning networks with GBLUP to effectively increase the prediction accuracy of complex traits [48].

### 4.2. Single-Step Genomic Best Linear Unbiased Prediction (ssGBLUP)

The ssGBLUP method is a genetic estimation method that integrates information from both breeds and genomes through a hybrid relationship matrix (H matrix) composed of a pedigree relationship matrix (A matrix) and a genomic relationship matrix (G matrix) [49]. This method enables the simultaneous use of data from animals with and without genotypes, resulting in more accurate genetic value predictions, even with low heritability, and can be performed in a single step [50]. The ssGBLUP method is suitable for predicting complex traits, such as the egg production performance study of the NCHU-G101 chicken line [44]. In addition, Zhu et al. [51] also found that ssGBLUP has higher genetic value accuracy than GBLUP, especially in growth and reproductive traits in Wenchang chickens. Previous studies confirmed that using genomic information improves the accuracy of genetic value predictions. However, compared to using pedigree information alone [52,53], the study by Makanjuola et al. [54] found that BLUP provided higher predictive values than ssGBLUP for fertility traits. A major limitation of ssGBLUP is its sensitivity to missing or low-quality data, as combining data from both genotyped and non-genotyped animals can reduce its accuracy if genotype or phenotype data are incomplete, or if there are differences in the genetic structure of the population. Proposed solutions include the use of accurate imputation methods, algorithmic improvements to increase efficiency, and the development of indirect genetic value (GEBV) methods. A study of 2.6 million Irish cattle with 500,000 genomes found that using such methods significantly improved the accuracy (correlation > 0.99), reduced dispersal and bias, and increased the reliability of the ssGBLUP model [55]. However, ssGBLUP is still limited in predicting traits controlled by major effect genes, which may require more advanced and flexible models, such as Bayesian or WGBLUP models, to overcome this limitation.

### 4.3. Bayesian Approaches

The Bayesian method uses statistical principles to calculate the GEBV by estimating the genetic value from SNP data through prior and posterior distributions using the Markov chain Monte Carlo (MCMC) process to provide highly accurate and flexible calculation results [56]. This method is popular for GS because it can handle differences in effects. The SNP and data incompleteness problems are better than those of GBLUP [57]. Bayesian analysis can yield better results than GBLUP for reproductive traits with moderate-to-large quantitative trait locus (QTL) effects, especially when using many SNPs. It also provides useful information regarding SNPs, such as estimating the impact size and variance, which helps with mapping. QTLs are utilized to enhance breeding and identify varieties with superior reproductive characteristics [58]. The Bayes A model is used when it is believed that all SNPs have an effect with a normal distribution of effects, whereas Bayes B is suitable for cases where some SNPs have a high impact, while many have no effect [40]. In a study by Wolc et al. [25], the Bayes model was used to determine whether SNPs (positions in the genome) affected approximately 1% of genetic traits, of which six traits were studied: sperm motility, sperm count, fertility using AI, fertility, fertile hatching, and the hatching of sets [59]. Therefore, Bayes C and Bayes D have been developed to calculate π values from real data, enabling the model to better cope with the larger number of SNPs and making genetic value predictions more accurate [60]. Therefore, choosing a model suitable for the genetic characteristics of a population is important. However, the Bayesian method also has disadvantages, such as high computational costs. It requires considerable processing power and a long calculation time because it requires repetitive calculation processes, such as MCMC or variational Bayes, as well as the optimization of hyperparameters. If the prior distributions are not carefully selected, they can cause model bias or slow convergence. Additionally, this method is sensitive to small sample sizes, as it requires extensive data to accurately estimate parameters, leading to potentially inaccurate genetic value predictions when datasets are limited. Finally, the results obtained using the Bayesian method are often posterior distributions, which can be challenging to interpret for those who do not have deep knowledge in this area [61]. However, their effectiveness depends greatly on the choice of prior distributions for model parameters, including marker effects and variance components. To help researchers select appropriate priors, we provided an explanation of commonly used prior distributions, such as normal, Student’s t, and Laplace priors, along with their advantages, based on the genetic architecture of the trait. For example, heavy-tailed priors (e.g., Student’s t) were useful for handling large-effect QTLs, while Gaussian priors were more suitable for traits with a polygenic inheritance pattern. To address computational challenges, several approaches have been proposed. Variational Bayesian techniques helped to reduce processing time [62], while parallel computing supported big data analysis [63]. Additionally, hierarchical Bayesian models were developed to optimize prior distribution selection [64].

### 4.4. Ridge Regression Genomic Best Linear Unbiased Prediction (RR-GBLUP)

RR-GBLUP is a GS method based on the idea that the effects of SNP markers are normally distributed, uncorrelated, and equally variable [65]. It is like GBLUP in terms of its concepts, but it uses ridge regression to make the model easier to understand. This enables the stable prediction of genetic values, especially in a manner controlled by many SNPs, each of which has little effect [66]. The advantages of RR-GBLUP include its ease of use and resistance to genetic information controlled by multiple genes (polygenic traits), which are considered polygenic for semen and heat tolerance [23,29]. This makes it a good choice for traits influenced by many SNPs with minimal effects. VanRaden [67] showed that RR-GBLUP is a viable method for quantifying traits in chickens, especially when a trait is influenced by multiple SNPs spread across the genome. In addition, Abdollahi-Arpanahi et al. [68] used RR-GBLUP to predict the egg-laying characteristics and egg quality of laying hens, and the predictive accuracy values were close to those of GBLUP, which showed that RR-GBLUP could be used as an effective choice for the genetic selection of these traits. However, RR-GBLUP had a major limitation. This method assumed that all SNP markers contributed equally to genetic variance, which was not ideal for traits influenced by major genes. In cases where a few loci had large effects, this method tended to overestimate the importance of many small-effect SNPs while underestimating the impact of major QTLs. As a result, it reduced the accuracy of genomic predictions for such traits. A possible solution was to use Bayesian models, such as Bayes B, which were better at detecting epistasis. Alternatively, multi-trait models, such as multi-trait GBLUP or Bayesian multi-trait models, could be applied to improve accuracy by utilizing the genetic relationships between different traits [40,59,69,70].

### 4.5. Weighted Genomic Best Linear Unbiased Prediction (WGBLUP)

WGBLUP is a development of GBLUP that adds weight to the G matrix so that each SNP influences the model differently based on its actual impact on the phenotype [71], instead of giving all animals the same weight, such as in GBLUP, which may not be accurate in predicting the genetic traits of interest that are controlled by a small but high-impact QTL or that are controlled by high-impact primary SNPs, such as growth, egg production, and disease resistance [58,72]. It can also reduce the effects of non-important SNPs, thereby stabilizing predictable genetic values. However, WGBLUP has a significant limitation, as adding weights to SNPs in the G matrix complicates the computation process and takes more time to analyze than the traditional GBLUP method. In addition, setting inappropriate initial weights may reduce the accuracy of the prediction results, especially when dealing with a large number of SNPs in large datasets, which makes the analysis a significant burden. One of the solutions that has been studied is to reduce the dimensionality of the SNP data to reduce the burden of the WGBLUP G matrix calculation, especially using the APY (Algorithm for Proven and Young) method, which can efficiently estimate the inverse of the G matrix in large datasets [73]. In addition, there are approaches to improve the SNP weight setting by using the iterative reweighting process, which uses data from GWAS analysis or Bayesian variable selection models to continuously update the weight values, which helps to initialize the appropriate weight values and increase the prediction accuracy. In addition, combining with the ssGBLUP model can integrate information from both genotypes and ancestry, reduce computation steps, and increase overall efficiency [47].

### 4.6. Multi-Trait Genomic Best Linear Unbiased Prediction (MTGBLUP)

The MTGBLUP model extends the GBLUP model to handle multiple features simultaneously. By relying on the genetic relationships between different traits to increase the accuracy of predicting the GEBV of animals, this method is suitable for breeding to improve many traits [74], such as semen mobility, concentration, and content. We considered the interrelated characteristics simultaneously. This made the selection of varieties more accurate than in single-style GBLUP [74,75]. For example, Zhang et al. [76] found that multi-trait models can provide more accurate predictions of genetic values. This is particularly true when the traits are genetically related. The selection of varieties can optimize growth and feed consumption without affecting the mortality rate. However, one problem with MTGBLUP is that it requires a large amount of high-quality data to estimate the covariance between traits [75]. The solution uses local genetic correlations (LGCs) to reduce the complexity of the covariance calculation between multiple traits [77]. In addition, Bayesian posterior-weighted GBLUP can add weights to high-impact SNPs based on Bayesian posterior covariance values. These methods not only reduce the computational complexity but also effectively improve the accuracy and efficiency of MTGBLUP in predicting heritability in multiple traits [78].

All models share common limitations in terms of the need for high-quality and complete data, complex processing, and sensitivity to missing or low-quality data. Solutions to these limitations may include the use of advanced processing technologies such as parallel computing to reduce processing time; the use of imputation techniques to fill in missing data, which improves the quality of the data; and the development of flexible models such as multi-task Bayesian learning models to accommodate the characteristics of data in each population [79]. In addition, the design of analytical methods that support complex relationships, such as epistatic models, can effectively reduce limitations and increase the accuracy of genetic trait predictions [69].

From the data in Table 1, the user can decide on the most suitable model by considering many factors. The first is the accuracy required to evaluate the characteristics of different models with different levels of accuracy. The second is the processing time, which affects the speed at which the results are obtained. The ssGBLUP model, which takes less time, might be a viable option, whereas the Bayesian model may be suitable when there is no rush but higher accuracy is required. In addition, available resources, such as processing power and the ability to handle specifications, affect the choice of the model. The optimal genomic selection model should be chosen based on specific traits of interest (e.g., production, fertility, health, or environmental adaptation), which aids breeders in making efficient and targeted breeding decisions.

## 5. Genome-Wide Association Study (GWAS)

A genome-wide association study (GWAS) is a useful method for finding genetic variations linked to quantitative traits. It does this by using genetic markers like single nucleotide polymorphisms (SNPs) to compare the number of alleles in groups with the desired genotype and control groups. Statistical models in GWASs set the confidence level of the result at *p* < 10^−8^, which reflects the reliability of the detected SNP associated with the phenotypic trait. GWASs have been used in the past to look at sperm quality and heat tolerance in chickens by finding important SNPs and genes (Table 2). For example, the LOXL1 gene affects how sperm moves after freezing [80], as well as the ENSGALG00000029931 gene and the PHF14 and ARID1B genes, which affect damaged seminal vesicles [81]. Zhang et al. [82] also indicated that the FAPP1, OSBPL6, SESTD1, and SSFA2 genes affected the number and size of sperm when they were exposed to extreme heat. These findings enhanced the understanding of the genetic basis controlling semen quality and response to heat. However, before these SNPs and genes could be applied in genetic selection, they needed to be validated in independent populations to confirm their effects on semen quality across various environments. Functional validation studies, such as gene knockout or knock-in experiments, helped to clearly establish the roles of these genes in biological processes [83,84]. Additionally, gene-by-environment (G × E) analysis allowed researchers to assess gene expression under different environmental conditions, enabling the identification of genotypes best suited for specific environments. Once validated, these SNPs could be incorporated into Marker-Assisted Selection (MAS) strategies to improve the accuracy of genetic selection. They could also be combined with genomic selection (GS) using models such as whole-genome best linear unbiased prediction (WGBLUP), which assigned greater weight to SNPs with large effects on phenotypes. Alternatively, Bayesian methods could be used to prioritize SNPs with direct biological significance. Integrating GWAS findings with GS not only enhanced the accuracy of genetic improvement but also facilitated the development of selection strategies that responded more effectively to environmental changes. Thus, these findings advanced the understanding of the genetic basis of semen quality and heat tolerance while supporting the development of sustainable genetic improvement strategies. Ultimately, this approach contributed to the increased productivity and enhanced adaptability of animals to changing environmental conditions.

## 6. Selection of Chicken Breeding Methods for Production Goals

Chicken breeding can be separated into various categories, each with a range of expenses and each being appropriate for various procedures. Breeding for distinct traits and production goals involves three primary groups: parent stock (PS), grandparent stock (GP), and great-grandparent stock (GGP). GGP are used in basic breeding to produce superior breeds in the future. High-precision methods such as pedigree-based selection and GS will help to make selection more accurate and efficient. This level of investment will be costly because it will require DNA testing and genetic data management technologies; however, it will be worthwhile in the long term because breeding at this level will assist in lowering overall production costs and increasing the efficiency of the following generation. Breeding that closely resembles actual production, such as choosing male or female chickens to produce eggs or meat, is known as GP. EBV and GS enable the selection of hens with favorable production traits, including growth and egg output. This investment will be moderately expensive, requiring the assessment of the animal’s performance and genetic information from both parents. However, it will be beneficial in the medium-term because it will speed up productivity improvements. Consequently, the field of genetics will continue to advance. PS are bred at the actual production level, as chickens are raised on commercial farms for meat or eggs. These farms frequently employ performance testing and phenotypic selection methods to examine external traits and actual production. Compared to GGP and GP, choosing breeds at this level is the least expensive because it does not require sophisticated technology, but still involves examining animal care and production. Therefore, it is cost-effective and suitable for short-term breeding in commercial production scenarios that require rapid output.

## 7. Conclusions

Despite their environmental adaptability, native chickens are significantly affected by increased HS, which in turn affects male fertility. Genetic improvement through GS and GWASs has proven effective in reducing the genetic selection time and cost, making it suitable for GGP chickens, which are a valuable breed. Breeders can use genomic testing to identify heat-tolerant and high-semen-quality traits, enhancing breeding plans with targeted equipment and skilled personnel.

## Figures and Tables

**Figure 1 animals-15-01096-f001:**
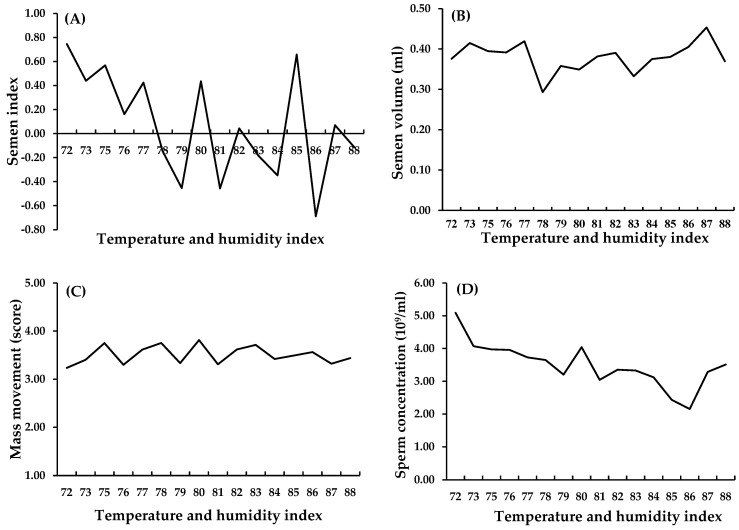
Semen index (**A**), semen volume (**B**), mass movement (**C**), and sperm concentration (**D**) across varying temperature and humidity indices.

**Figure 2 animals-15-01096-f002:**
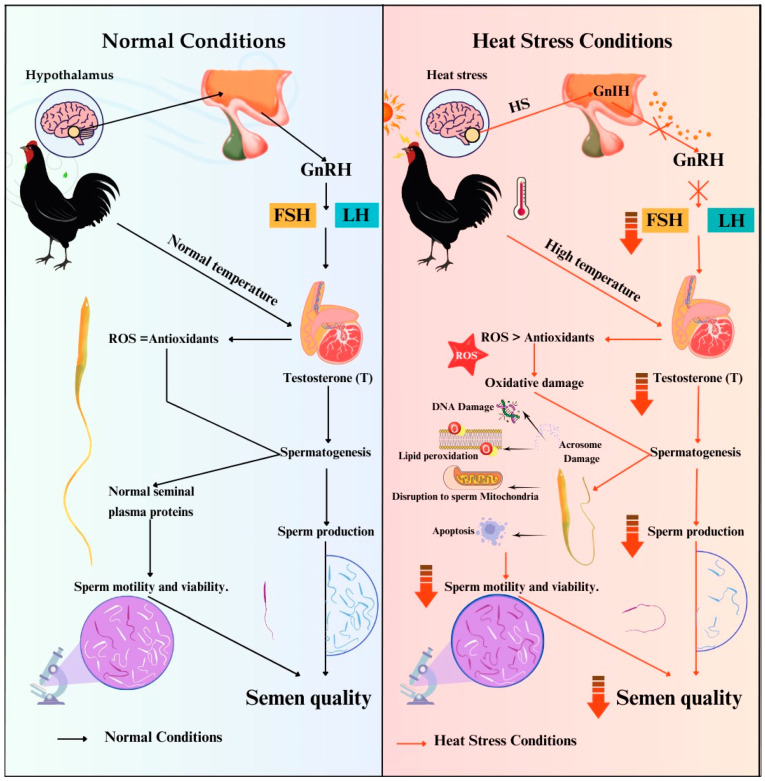
Hormonal regulation, semen production, and sperm quality in roosters under normal and heat stress conditions.

**Figure 3 animals-15-01096-f003:**
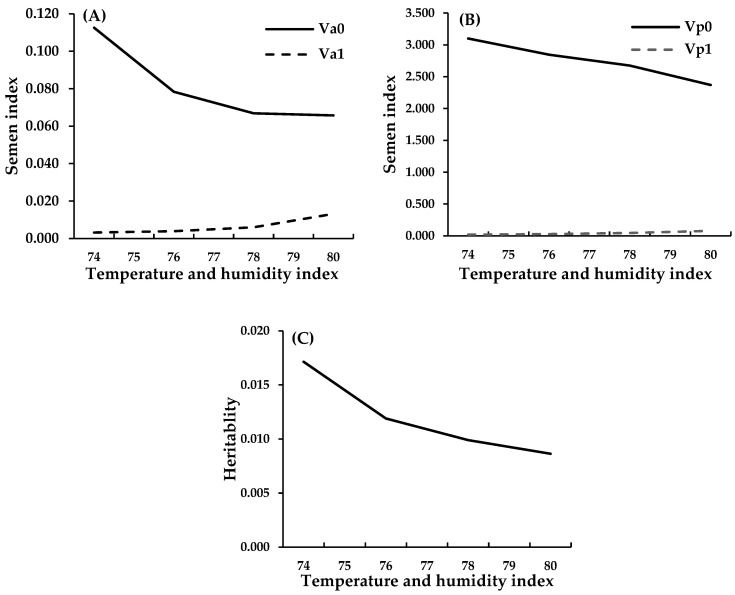
The additive genetic variance estimated without heat tolerance (intercept, Va0, solid line) and with heat tolerance (slope, Va1, dashed line) (**A**). Permanent environmental variance without HS (intercept, Vp0, solid line) and with heat tolerance (slope, Vp1, dashed line) (**B**) and heritability (**C**).

**Table 1 animals-15-01096-t001:** Comparison of genomic prediction models.

Model	Trait	Process (Step)	Accuracy (%)	Time (Hour)	Program	Limit	Published
GBLUP	Production	2	~70–85%	3–5	ASReml BLUPF90	Assumes normal distribution	Yes
	Fertility						
ssGBLUP	Fertility	1	~75–90%	1–2	ssGBLUP ASReml	Requires accurate pedigree data, higher RAM usage with large datasets	Yes
	Diseases						
	Health						
	Environmental						
Bayesian	Quality	2–3	~80–95%	12–48	BayesR BGLR	Slowest computation time, prior SNP distribution setup needed	No
	Fertility						
	Health						
	Health						
RR-GBLUP	Production	2	~65–80%	3–5	rrBLUP	Assumes no linkage disequilibrium	No
WGBLUP	Fertility	2	~75–90%	5–8	ASReml BLUPF90	Needs GWAS data for SNP weighting	No
	Diseases						
	Health						
	Environmental						
MTGBLUP	Production	2	~80–95%	5–8	MTG2 BLUPF90	Longer computation, requires ample multi-trait data	No
	Quality						
	Fertility						
	Environmental						

**Table 2 animals-15-01096-t002:** Genome-wide association analysis of rooster fertility traits.

SNP Number	Gene	SNPs	Chromosome	Position	Trait	Reference
60 K	LOXL1	rs15557972	10	9810123	Sperm motility	[80]
	ENSGALG00000052127	rs15751385	6	34380465	Sperm motility	
600 K	TRPC1	Affx-51823443	9	10568531	Sperm concentration	[85]
		Affx-51823444	9	10568594	Sperm concentration	
	SLC9A9	Affx-51824235	9	10856343	Sperm concentration	
		Affx-51824375	9	10909735	Sperm concentration	
	CUL3	Affx-51873906	9	8433286	Sperm concentration	
	MTF1	Affx-51092352	23	3563615	Sperm concentration	
		Affx-51092326	23	3557077	Sperm concentration	
600 K	PHF14	AX-76063628	2	26182792	Sperm membrane	[86]
	ARID1B	AX-76495998	3	51262693	Sperm membrane	
55 K	FAPP1, OSBPL6,	Not specified	7	13820000–16120000	Semen volume	[82]
	SESTD1, SSFA2					
600 K	ENSGALG00000029931	Not specified	Not specified	168850183	Sperm motility	[81]
	KDELR3	AX-75466971	1	50898160	Sperm respiration	
	DDX17	AX-75466971	1	50898160	Sperm respiration	
	DMD	AX-75221789	1	116157001	Sperm respiration	
	CDKL5	AX-75231769	1	122024645	Sperm respiration	
	DGAT2	AX-75397985	1	196966714	Sperm respiration	
	ST18	AX-80992139	2	109830505	Sperm respiration	
	FAM150A	AX-80992139	2	109830505	Sperm respiration	
	DIAPH2	AX-80778510	4	5664389	Sperm respiration	
	MTMR7	AX-76705102	4	63101468	Sperm respiration	
	NAV2	AX-76788932	5	1970758	Sperm respiration	
	RAG2	AX-76791651	5	20089359	Sperm respiration	
	PDE11A	AX-76986124	7	15619919	Sperm respiration	
	IFT70A	AX-76986304	7	15687930	Sperm respiration	
	AGPS	AX-76986304	7	15687930	Sperm respiration	
	WDFY1	AX-77181439	9	8463525	Sperm respiration	
	DEPDC5	AX-75848147	15	9143016	Sperm respiration	
	TSC1	AX-75873724	17	7048201	Sperm respiration	
	CASZ1	AX-76244713	21	3983187	Sperm respiration	
	PLEKHM2	AX-76245698	21	4201372	Sperm respiration	

## Data Availability

The data presented in this study are available upon reasonable request from the Network Center for Animal Breeding and Omics Research, Faculty of Agriculture, Khon Kaen University, Thailand.

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
