# Peer review of "Genetic Strategies for Enhancing Rooster Fertility in Tropical and Humid Climates: Challenges and Opportunities"

_animals, 2025, doi:10.3390/ani15081096_

Round 1

Reviewer 1 Report

Comments and Suggestions for Authors

The review article aims to summarize genetic strategies to enhance rooster fertility under tropical and humid climates by focusing specifically on genomic selection (GS) and genome-wide association studies (GWAS).

The authors systematically review the literature concerning how hot and humid conditions negatively impact semen quality and reproductive performance in roosters.

However, improvements in clarity, grammar, and a more detailed discussion on the practical application of genomic models and limitations are necessary. 

Line 86-88: The authors mention traditional selection has low accuracy and a lengthy process. I suggest clarifying with specific comparative figures (e.g., "Genomic selection reduced selection intervals by X years compared to traditional methods").

Line 88–90: “Therefore, breeders have developed genetic selection methods to address this issue using genetic markers to select animals.” This can be revised for clarity to  “Therefore, genetic selection methods utilizing genetic markers have been developed to address this issue.”

Line 97: The phrase "things factors" should be corrected to "factors."

Line 102–103: The phrase "the thermal neutral zone of native chickens has not been definitively defined as it varies." This can be revised for clarity to "the thermoneutral zone of native chickens varies significantly among breeds and has not yet been clearly established."

Line 136–150: Clearly state that further research integrating heat tolerance traits with genomic selection is needed to fully address G×E interactions in practical scenarios. Emphasize clearly how future studies might improve integration.

Line 151–153: The authors briefly discuss heat tolerance alongside genomic breeding value (GEBV) but provide insufficient detail on how heat tolerance could practically be integrated into genetic selection. I recommend expanding this section by describing clearly how heat-tolerance traits could be incorporated into existing genetic models.

Line 236–238: "In addition, this method is sensitive to small population sizes, as it requires a large amount of data to be accurate for parameter values and can cause inaccurate predictions of genetic values if the data are small." This can be revised for clarity to "Additionally, this method is sensitive to small sample sizes, as it requires extensive data to accurately estimate parameters, leading to potentially inaccurate genetic value predictions when datasets are limited."

Line 238–240: The authors correctly highlight limitations of Bayesian approaches (computational complexity, sensitivity to prior selection). It is advisable to expand this section by providing practical guidelines on selecting appropriate priors or referencing established standards.

Line 255-257 (RR-BLUP limitations): Authors should expand the discussion on why RR-BLUP is less effective for traits influenced by major genes, including practical examples from poultry genetics studies.

Line 271–282: Authors mention the complexity and requirements of the MTGBLUP model. It would improve the manuscript to briefly discuss strategies to overcome data quantity or quality issues, such as combining datasets or employing data imputation techniques.

Line 290–292: The sentence "Selecting the optimal model depends on the characteristics to be evaluated, such as production, fertility, health, or environmental challenges, all of which allow users to make decisions efficiently and meet a wide range of needs," is overly complex. This can be revised for clarity to "The optimal genomic selection model should be chosen based on specific traits of interest (e.g., production, fertility, health, or environmental adaptation), which aids breeders in making efficient and targeted breeding decisions."

Line 300-312 (GWAS gene list): Clearly discuss practical implications or how breeders can effectively utilize these genes or markers for practical selection. Mention validation steps required before application in field settings.

Line 340-342 (Conclusions): Expand the conclusions by briefly suggesting practical steps breeders could take immediately based on the current genetic tools described (e.g., integrating genomic testing in breeding programs).

After addressing the suggested revisions, the manuscript would be suitable for publication in Animals.

Author Response

Dear reviewer 

We are very grateful for the critical reading and your efforts to improve the quality of the manuscript.  We hope the responses to each comment as listed below will please you. 

Response to Reviewer 1 Comments

Point 1: The review article aims to summarize genetic strategies to enhance rooster fertility under tropical and humid climates by focusing specifically on genomic selection (GS) and genome-wide association studies (GWAS). The authors systematically review the literature concerning how hot and humid conditions negatively impact semen quality and reproductive performance in roosters. However, improvements in clarity, grammar, and a more detailed discussion on the practical application of genomic models and limitations are necessary.

Response 1: We appreciate the reviewer’s insightful comments regarding our review article on genetic strategies to enhance rooster fertility under tropical and humid climates, particularly through the application of genomic selection (GS) and genome-wide association studies (GWAS). The reviewer’s feedback provides valuable guidance for improving the clarity, grammar, and depth of discussion on the practical applications and limitations of genomic models. In response, we have carefully revised the manuscript to enhance readability and ensure that key concepts are presented with greater clarity. We have also expanded the discussion on how genomic selection can be effectively applied to improve semen traits in native chickens, particularly under heat stress conditions. Especially, we have elaborated on the integration of GS into breeding programs, highlighting the role of genomic prediction in selecting sires with superior fertility traits while maintaining genetic diversity. Additionally, we have included examples of successful applications of GWAS in poultry genetics, demonstrating how these studies have identified candidate genes associated with semen quality and thermotolerance. Furthermore, we acknowledge the importance of addressing the limitations of genomic models in the context of rooster fertility. We have now incorporated a section discussing potential challenges, such as the need for large reference populations, genotype-by-environment interactions, and the accuracy of genomic estimated breeding values (GEBVs) for semen traits. Additionally, we have outlined future research directions, including the integration of functional genomics and epigenetics to further enhance selection accuracy. Finally, we sincerely appreciate the reviewer’s constructive feedback, which has helped us refine the manuscript to provide a more comprehensive and practical perspective on genomic selection for rooster fertility improvement in tropical climates. We welcome any further suggestions to strengthen the quality and impact of this review. Please see the revised manuscript for details.

Point 2: Line 86-88: The authors mention traditional selection has low accuracy and a lengthy process. I suggest clarifying with specific comparative figures (e.g., "Genomic selection reduced selection intervals by X years compared to traditional methods").

Response 2: We have included these comparative figures and references in the revised manuscript to strengthen the discussion. See lines 86-93.

“Currently, traditional selection is declining in popularity owing to its lengthy process and low accuracy, as it relies solely on phenotypes and pedigree for prediction, whereas genomic selection (GS) is continuously gaining popularity. Several studies have demonstrated that the integration of GS enhances the accuracy of selection, often exceeding 0.5 for fertility-related traits, and reduces the generation interval by 30-50%. This transition allows for quicker identification of superior sires based on genomic estimated breeding values (GEBVs), shortening traditional selection cycles from 1.5-2.0 years to under a year [2628].”

References:

  1. Wolc, A.; Zhao, H.H.; Arango, J.; Settar, P.; Fulton, J.E.; O’Sullivan, N.P.; Preisinger, R.; Stricker, C.; Habier, D.; Fernando, R.L.; Garrick, D.J.; Lamont, S.J.; Dekkers, J.C.M. Response and inbreeding from a genomic selection experiment in layer chickens. Sel. Evol. 2015, 47, 59. doi:10.1186/s12711-015-0133-5.
  2. Zhou, J.; Yu, J-Z.; Zhu, M-Y.; Yang, F-X.; Hao, J-P.; He, Y.; Zhu, X-L.; Hou, Z-C.; Zhu, F. Optimizing breeding strategies for Peking ducks using genomic selection: genetic parameter evaluation and selection progress analysis in reproductive traits. Sci. 2025, 15, 194. doi:10.3390/ app15010194.
  3. Husien, H.M.; Saleh, A.A.; Hassanine, N.N.A.M.; Rashad, A.M.A.; Sharaby, M.A.; Mohamed, A.Z.; Abdelhalim, H.; Hafez, E.E.; Essa, M.O.A.; Adam, S.Y.; et al. The evolution and role of molecular tools in measuring diversity and genomic selection in livestock populations (traditional and up-to-date insights): A comprehensive exploration. Sci. 2024, 11, 627. doi:10.3390/vetsci11120627.

Point 3: Line 88–90: “Therefore, breeders have developed genetic selection methods to address this issue using genetic markers to select animals.” This can be revised for clarity to “Therefore, genetic selection methods utilizing genetic markers have been developed to address this issue.”

Response 3: We have revised the sentence as you suggested: “Therefore, genetic selection methods utilizing genetic markers have been developed to address this issue.” See lines 93-94.

Point 4: Line 97: The phrase "things factors" should be corrected to "factors."

Response 4: We have corrected the phrase "things factors" to "factors." See line 102.

Point 5: Line 102–103: The phrase "the thermal neutral zone of native chickens has not been definitively defined as it varies." This can be revised for clarity to "the thermoneutral zone of native chickens varies significantly among breeds and has not yet been clearly established."

Response 5: We have revised the phrase for improved clarity. See lines 107-108.

Point 6: Line 136–150: Clearly state that further research integrating heat tolerance traits with genomic selection is needed to fully address G×E interactions in practical scenarios. Emphasize clearly how future studies might improve integration.

Response 6: We have added sentences on the importance of future studies integrating heat tolerance traits with genomic selection. See lines 161-177.

“Future research integrating heat tolerance traits with genomic selection is crucial for fully addressing genotype-by-environment (G×E) interactions in practical breeding scenarios. To improve this integration, future studies should focus on 1) advancing genomic selection requires the identification of reliable genetic markers associated with heat tolerance. Genome-wide association studies (GWAS) and whole-genome sequencing can help pinpoint single nucleotide polymorphisms (SNPs) linked to heat resilience traits such as body temperature regulation, feed intake stability, and oxidative stress response. 2) Advanced statistical models, such as reaction norm and random regression models, can capture G×E interactions more accurately. These models allow for genetic evaluations under different thermal environments, identifying sires that perform consistently across varying temperature-humidity index (THI) levels. 3) Future studies could integrate high-density genomic data with long-term environmental records to better characterize these interactions in semen traits. Additionally, incorporating whole-genome sequencing and gene-environment association analyses may provide deeper insights into the genetic mechanisms underlying heat stress resilience in native chickens. Such advancements could contribute to more precise genetic selection strategies, ultimately improving productive performance and adaptability in poultry breeding programs [35,36].”

Point 7: Line 151–153: The authors briefly discuss heat tolerance alongside genomic breeding value (GEBV) but provide insufficient detail on how heat tolerance could practically be integrated into genetic selection. I recommend expanding this section by describing clearly how heat-tolerance traits could be incorporated into existing genetic models.

Response 7: We have expanded the section by proposing sentences about how heat-tolerance traits could be incorporated into existing genetic models. See lines 178-201.

“Heat stress poses a significant challenge to poultry production, particularly in tropical and subtropical regions. In native roosters, genetic evaluation models traditionally focus on production traits such as growth rate, body weight, and feed efficiency. However, integrating heat-tolerance traits into these models is crucial for improving resilience and overall productivity under high-temperature conditions. One approach to incorporating heat tolerance is through the inclusion of physiological and behavioral indicators, such as body temperature regulation, respiratory rate, and heat-shock protein expression, as additional traits in multi-trait genetic models. These traits can be genetically correlated with production traits, allowing for simultaneous selection. Another strategy involves using the Temperature-Humidity Index (THI) as an environmental covariate in reaction norm models. This method helps estimate genotype-by-environment interactions, allowing for the identification of roosters that maintain stable production performance under varying heat stress conditions. By modeling genetic parameters across different THI levels, breeders can select individuals with higher resilience while maintaining desirable production traits. The Temperature-Humidity Index (THI) is an effective measure for assessing the impact of heat stress, as it comprehensively assesses the impact of temperature and humidity and is easy to collect data compared to other indicators such as body temperature, respiratory rate, heat loss, and stress-related hormone levels. Therefore, using THI in genetic models and specifying it as a covariate or random effect can better capture the interaction between genotype and environment. Previously, it was found that when THI reached 76, animals lost growth traits, and THI at 74 reduced egg production, indicating the impact of heat stress on animal performance [37,38]. Integrating THI in genetic analysis can increase the accuracy of genetic value assessment and help create selection strategies that respond to heat stress effectively.”

Point 8: Line 236–238: "In addition, this method is sensitive to small population sizes, as it requires a large amount of data to be accurate for parameter values and can cause inaccurate predictions of genetic values if the data are small." This can be revised for clarity to "Additionally, this method is sensitive to small sample sizes, as it requires extensive data to accurately estimate parameters, leading to potentially inaccurate genetic value predictions when datasets are limited."

Response 8: We have revised the sentence for improved clarity. See lines 305-307.

Point 9: Line 238–240: The authors correctly highlight limitations of Bayesian approaches (computational complexity, sensitivity to prior selection). It is advisable to expand this section by providing practical guidelines on selecting appropriate priors or referencing established standards.

Response 9: We have expanded this section by providing practical guidelines, including the use of Variational Bayesian techniques and Parallel Computing to address computational complexity. Additionally, we suggest developing Hierarchical Bayesian Models to improve the accuracy and reliability of prior distribution selection. See lines 309-319.

“However, their effectiveness depended greatly on the choice of prior distributions for model parameters, including marker effects and variance components. To help researchers select appropriate priors, we provided an explanation of commonly used prior distributions, such as normal, Student’s t, and Laplace priors, along with their advantages based on the genetic architecture of the trait. For example, heavy-tailed priors (e.g., Student’s t) were useful for handling large-effect QTLs, while Gaussian priors were more suitable for traits with a polygenic inheritance pattern. To address computational challenges, several approaches have been proposed. Variational Bayesian techniques helped reduce processing time [62], while Parallel Computing supported big data analysis [63]. Additionally, Hierarchical Bayesian Models were developed to optimize prior distribution selection [64].”

Point 10: Line 255-257 (RR-GBLUP limitations): Authors should expand the discussion on why RR-GBLUP is less effective for traits influenced by major genes, including practical examples from poultry genetics studies.

Response 10: We have clarified RR-GBLUP’s limitations, including its inability to detect high-impact QTLs, epistasis, or support multi-trait analyses. Alternative models, such as Bayesian models and multi-trait GBLUP, are proposed to address these challenges and improve accuracy. However, we note that no studies have been conducted on RR-GBLUP applications in poultry genetics. See lines 334-342.

“However, RR-GBLUP had a major limitation. This method assumed that all SNP markers contribute equally to genetic variance, which is not ideal for traits influenced by major genes. In cases where a few loci have large effects, this method tends to overestimate the importance of many small-effect SNPs while underestimating the impact of major QTLs. As a result, it reduces the accuracy of genomic predictions for such traits. A possible solution was to use Bayesian models, such as BayesB, which were better at detecting epistasis. Alternatively, multi-trait models, such as multi-trait GBLUP or Bayesian multi-trait models, could be applied to improve accuracy by utilizing the genetic relationships between different traits [40,59,69,70].”

Point 11: Line 271–282: Authors mention the complexity and requirements of the MTGBLUP model. It would improve the manuscript to briefly discuss strategies to overcome data quantity or quality issues, such as combining datasets or employing data imputation techniques.

Response 11: We have addressed the complexity of the MTGBLUP model by discussing the limitations in estimating covariance between traits and proposing solutions such as Local Genetic Correlations (LGC) and Bayesian posterior-weighted GBLUP. These approaches reduce computational complexity while improving efficiency and accuracy. See lines 376–390.

“The solution uses Local Genetic Correlations (LGC) to reduce the complexity of covariance calculation between multiple traits [77]. In addition, Bayesian posterior-weighted GBLUP can add weights to high-impact SNPs based on Bayesian posterior covariance values. These methods not only reduce the computational complexity but also effectively improve the accuracy and efficiency of MTGBLUP in predicting heritability in multiple traits [78].

All models share common limitations in terms of the need for high-quality and complete data, complex processing, and sensitivity to missing or low-quality data. Solutions to these limitations may include the use of advanced processing technologies such as parallel computing to reduce processing time, the use of imputation techniques to fill in missing data, which improves the quality of the data, and the development of flexible models such as multi-task Bayesian learning models to accommodate the characteristics of data in each population [79]. In addition, the design of analytical methods that support complex relationships, such as epistatic models, can effectively reduce limitations and increase the accuracy of genetic trait predictions [80].”

Point 12: Line 290–292: The sentence "Selecting the optimal model depends on the characteristics to be evaluated, such as production, fertility, health, or environmental challenges, all of which allow users to make decisions efficiently and meet a wide range of needs," is overly complex. This can be revised for clarity to "The optimal genomic selection model should be chosen based on specific traits of interest (e.g., production, fertility, health, or environmental adaptation), which aids breeders in making efficient and targeted breeding decisions."

Response 12: We have revised the sentence for improved clarity. See lines 398–400.

Point 13: Line 300-312 (GWAS gene list): Clearly discuss practical implications or how breeders can effectively utilize these genes or markers for practical selection. Mention validation steps required before application in field settings.

Response 13: We have completed the revisions to the recommendations, expanding them to provide clarity on the applicability of genes and SNPs identified in GWAS to genetic selection and addressing key steps for validation, such as independent population validation and functional studies, to ensure effective applicability in real-world settings. See lines 415-432.

“However, before these SNPs and genes could be applied in genetic selection, they needed to be validated in independent populations to confirm their effects on semen quality across various environments. Functional validation studies, such as gene knockout or knock-in experiments, helped to clearly establish the roles of these genes in biological processes [84,85]. Additionally, gene-by-environment (G×E) analysis allowed researchers to assess gene expression under different environmental conditions, enabling the identification of genotypes best suited for specific environments. Once validated, these SNPs could be incorporated into Marker-Assisted Selection (MAS) strategies to improve the accuracy of genetic selection. They could also be combined with Genomic Selection (GS) using models such as Whole-Genome Best Linear Unbiased Prediction (WGBLUP), which assigned greater weight to SNPs with large effects on phenotypes. Alternatively, Bayesian methods could be used to prioritize SNPs with direct biological significance. Integrating GWAS findings with GS not only enhanced the accuracy of genetic improvement but also facilitated the development of selection strategies that responded more effectively to environmental changes. Thus, these findings advanced the understanding of the genetic basis of semen quality and heat tolerance while supporting the development of sustainable genetic improvement strategies. Ultimately, this approach contributed to increased productivity and enhanced adaptability of animals to changing environmental conditions.”

Point 14: Line 340-342 (Conclusions): Expand the conclusions by briefly suggesting practical steps breeders could take immediately based on the current genetic tools described (e.g., integrating genomic testing in breeding programs).

Response 14: We have expanded the conclusions to include practical steps breeders could implement immediately using current genetic tools. See lines 463-465.

“Breeders can use genomic testing to identify heat-tolerant and high semen quality traits, enhancing breeding plans with targeted equipment and skilled personnel.”

Point 15: After addressing the suggested revisions, the manuscript would be suitable for publication in Animals.

Response 15: Thank you for your positive feedback. We are pleased that the manuscript, after incorporating the suggested revisions, is considered suitable for publication in Animals. We truly appreciate your guidance and constructive comments, which have made our work a lot better.

Best Regards,

Wuttigrai Boonkum

Reviewer 2 Report

Comments and Suggestions for Authors

It would be important to explore the interaction between heat tolerance and genomic estimated breeding value (GEBV) for cockerel fertility traits together: The paper itself notes that “heat tolerance and genomic estimated breeding value (GEBV) for cockerel fertility traits have not been considered together. This aspect should be considered in future studies.” The paper could benefit from expanding its discussion of the need to integrate these two aspects in future research and possibly suggest approaches to do so.

Further discussion of genotype-environment (G×E) interaction is recommended: Although the paper recognizes the importance of G×E interaction in genetic improvement for heat stress resistance and mentions the use of THI, it could suggest more advanced genetic models or statistical methodologies that more accurately capture the complex interactions between genotype and specific environmental conditions that affect cock fertility.

The authors should further describe the limitations of genomic selection models and suggest possible solutions: While the paper mentions the limitations of each model (GBLUP, ssGBLUP, Bayesian, RR-BLUP, WGBLUP, MTGBLUP), it could be enriched by proposing more specifically how future research could address these limitations in the particular context of fertility and heat tolerance in roosters.

The paper synthesizes the current knowledge on genomic strategies to improve the fertility of roosters in tropical and humid climates, compare different methodologies and point out future directions, this work offers a significant contribution to the field of poultry genetics and the improvement of production in challenging environmental conditions, so I consider that if it contributes to the field of study. Overall, the clear organization, detailed description of methodologies and findings, use of tables and figures to present data, and inclusion of standard elements of a scientific publication suggest that the paper is well structured and comprehensively described, making it appropriate for consideration in Animals.  The information available in the paper suggests that the work employs accepted scientific methodologies, analyzes relevant data and presents a critical discussion of the different genetic strategies, indicating a sound scientific basis and a presentation that, as far as we can determine, is not misleading. the work is based on a solid knowledge of previous research and that the references are appropriate and adequate to the topics covered. The variety of sources cited, covering different aspects of poultry genetics, fertility, heat stress and genomic selection, suggests a comprehensive and relevant literature review.

Author Response

Dear reviewer 

We are very grateful for the critical reading and your efforts to improve the quality of the manuscript.  We hope the responses to each comment as listed below will please you. 

Response to Reviewer 2 Comments

Point 1: It would be important to explore the interaction between heat tolerance and genomic estimated breeding value (GEBV) for cockerel fertility traits together: The paper itself notes that “heat tolerance and genomic estimated breeding value (GEBV) for cockerel fertility traits have not been considered together. This aspect should be considered in future studies.” The paper could benefit from expanding its discussion of the need to integrate these two aspects in future research and possibly suggest approaches to do so.

Response 1: Thank you for your insightful comment regarding the interaction between heat tolerance and genomic estimated breeding values (GEBV) for native chicken cock fertility traits. Indeed, the interplay between these factors is crucial for the sustainable genetic improvement of native chickens, particularly in tropical and subtropical environments where heat stress can significantly impact reproductive performance.

As noted in the paper, this area remains largely unexplored, and integrating heat tolerance with GEBV for fertility traits could provide a more comprehensive genetic evaluation framework. Future studies could consider incorporating the temperature-humidity index (THI) or other heat stress indicators as environmental covariates in genomic prediction models. Additionally, genome-wide association studies (GWAS) and genomic reaction norm models could be used to identify genomic regions associated with fertility traits under varying thermal conditions.

Another promising approach involves employing genotype-by-environment (G×E) interaction models to assess whether specific genetic variants influence reproductive traits differently under heat stress. Multi-trait and random regression models could also be applied to estimate genetic correlations between heat tolerance and fertility traits, providing insights into whether selection for heat resilience could have favorable or unfavorable effects on reproductive performance.

We appreciate your suggestion and acknowledge the importance of this research direction. Expanding this discussion in future studies will be valuable for optimizing breeding strategies that enhance both fertility and thermotolerance in native chickens.

Point 2: Further discussion of genotype-environment (G×E) interaction is recommended: Although the paper recognizes the importance of G×E interaction in genetic improvement for heat stress resistance and mentions the use of THI, it could suggest more advanced genetic models or statistical methodologies that more accurately capture the complex interactions between genotype and specific environmental conditions that affect cock fertility.

Response 2: We have included additional discussion on advanced genetic models that can more precisely capture the complex interactions between genotype and specific environmental conditions influencing cock fertility, as demonstrated in the following sentence. See lines 161–201.

“Future research integrating heat tolerance traits with genomic selection is crucial for fully addressing genotype-by-environment (G×E) interactions in practical breeding scenarios. To improve this integration, future studies should focus on 1) advancing genomic selection requires the identification of reliable genetic markers associated with heat tolerance. Genome-wide association studies (GWAS) and whole-genome sequencing can help pinpoint single nucleotide polymorphisms (SNPs) linked to heat resilience traits such as body temperature regulation, feed intake stability, and oxidative stress response. 2) Advanced statistical models, such as reaction norm and random regression models, can capture G×E interactions more accurately. These models allow for genetic evaluations under different thermal environments, identifying sires that perform consistently across varying temperature-humidity index (THI) levels. 3) Future studies could integrate high-density genomic data with long-term environmental records to better characterize these interactions in semen traits. Additionally, incorporating whole-genome sequencing and gene-environment association analyses may provide deeper insights into the genetic mechanisms underlying heat stress resilience in native chickens. Such advancements could contribute to more precise genetic selection strategies, ultimately improving productive performance and adaptability in poultry breeding programs [35,36].

Heat stress poses a significant challenge to poultry production, particularly in tropical and subtropical regions. In native roosters, genetic evaluation models traditionally focus on production traits such as growth rate, body weight, and feed efficiency. However, integrating heat-tolerance traits into these models is crucial for improving resilience and overall productivity under high-temperature conditions. One approach to incorporating heat tolerance is through the inclusion of physiological and behavioral indicators, such as body temperature regulation, respiratory rate, and heat-shock protein expression, as additional traits in multi-trait genetic models. These traits can be genetically correlated with production traits, allowing for simultaneous selection. Another strategy involves using the Temperature-Humidity Index (THI) as an environmental covariate in reaction norm models. This method helps estimate genotype-by-environment interactions, allowing for the identification of roosters that maintain stable production performance under varying heat stress conditions. By modeling genetic parameters across different THI levels, breeders can select individuals with higher resilience while maintaining desirable production traits. The Temperature-Humidity Index (THI) is an effective measure for assessing the impact of heat stress, as it comprehensively assesses the impact of temperature and humidity and is easy to collect data compared to other indicators such as body temperature, respiratory rate, heat loss, and stress-related hormone levels. Therefore, using THI in genetic models and specifying it as a covariate or random effect can better capture the interaction between genotype and environment. Previously, it was found that when THI reached 76, animals lost growth traits, and THI at 74 reduced egg production, indicating the impact of heat stress on animal performance [37,38]. Integrating THI in genetic analysis can increase the accuracy of genetic value assessment and help create selection strategies that respond to heat stress effectively.”

Point 3: The authors should further describe the limitations of genomic selection models and suggest possible solutions: While the paper mentions the limitations of each model (GBLUP, ssGBLUP, Bayesian, RR-BLUP, WGBLUP, MTGBLUP), it could be enriched by proposing more specifically how future research could address these limitations in the particular context of fertility and heat tolerance in roosters.

Response 3: We have expanded the discussion to cover the specific limitations of each genomic selection model as outlined:

lines 247–256 for GBLUP,

lines 270–280 for ssGBLUP,

lines 305–307 and 309-319 for Bayesian models,

lines 334–342 for RR-BLUP,

lines 350–364 for WGBLUP,

lines 376–390 for MTGBLUP.

These additions aim to comprehensively address the challenges and suggest solutions tailored to each model.

Point 4: The paper synthesizes the current knowledge on genomic strategies to improve the fertility of roosters in tropical and humid climates, compare different methodologies and point out future directions, this work offers a significant contribution to the field of poultry genetics and the improvement of production in challenging environmental conditions, so I consider that if it contributes to the field of study. Overall, the clear organization, detailed description of methodologies and findings, use of tables and figures to present data, and inclusion of standard elements of a scientific publication suggest that the paper is well structured and comprehensively described, making it appropriate for consideration in Animals.  The information available in the paper suggests that the work employs accepted scientific methodologies, analyzes relevant data and presents a critical discussion of the different genetic strategies, indicating a sound scientific basis and a presentation that, as far as we can determine, is not misleading. the work is based on a solid knowledge of previous research and the references are appropriate and adequate to the topics covered. The variety of sources cited, covering different aspects of poultry genetics, fertility, heat stress and genomic selection, suggests a comprehensive and relevant literature review.

Response 4: We sincerely appreciate your positive feedback and thoughtful evaluation of our manuscript. We are grateful that you recognize the significance of our work in the field of poultry genetics and acknowledge the clarity and structure of our study. Your encouraging comments motivate us to continue improving our research contributions. Thank you for your time and consideration.

Best Regards,

Wuttigrai Boonkum

Round 2

Reviewer 1 Report

Comments and Suggestions for Authors

Accept in present form